# Mechanosensation and Mechanotransduction by Lymphatic Endothelial Cells Act as Important Regulators of Lymphatic Development and Function

**DOI:** 10.3390/ijms22083955

**Published:** 2021-04-12

**Authors:** László Bálint, Zoltán Jakus

**Affiliations:** Department of Physiology, Semmelweis University School of Medicine, 1094 Budapest, Hungary; balint.laszlo@med.semmelweis-univ.hu

**Keywords:** lymphatics, lymphatic development, lymphatic function, mechanical forces, mechanosensation, mechanotransduction, lymphatic endothelial cell, signaling pathways

## Abstract

Our understanding of the function and development of the lymphatic system is expanding rapidly due to the identification of specific molecular markers and the availability of novel genetic approaches. In connection, it has been demonstrated that mechanical forces contribute to the endothelial cell fate commitment and play a critical role in influencing lymphatic endothelial cell shape and alignment by promoting sprouting, development, maturation of the lymphatic network, and coordinating lymphatic valve morphogenesis and the stabilization of lymphatic valves. However, the mechanosignaling and mechanotransduction pathways involved in these processes are poorly understood. Here, we provide an overview of the impact of mechanical forces on lymphatics and summarize the current understanding of the molecular mechanisms involved in the mechanosensation and mechanotransduction by lymphatic endothelial cells. We also discuss how these mechanosensitive pathways affect endothelial cell fate and regulate lymphatic development and function. A better understanding of these mechanisms may provide a deeper insight into the pathophysiology of various diseases associated with impaired lymphatic function, such as lymphedema and may eventually lead to the discovery of novel therapeutic targets for these conditions.

## 1. Introduction

Two vascular networks are present in mammals: the blood and the lymphatic systems. The blood vasculature is a circulatory vessel system that is essential for the transportation of respiratory gases, nutrients, and signaling molecules. Lymphatic vessels form a blind-ended, linear vessel system that is distinguished from the blood vasculature in its structure, function, and development, as reviewed previously by many investigators [1,2,3,4,5,6,7,8,9,10,11,12].

Both vessel systems are comprised of endothelial cells (EC), which are surrounded by smooth muscle cells, pericytes, and basal membrane in some vascular beds. ECs are differentiated to blood endothelial cells (BEC) in the blood vessel network and lymphatic endothelial cells (LEC) in the lymphatic vasculature. Initial lymphatics are comprised of loosely connected lymphatic endothelial cells forming button-like junctions [13,14], which allow the lymphatic capillaries to take up and transport fluid, macromolecules, and cells from the interstitial compartment. Lymph is transported from the initial lymphatic capillaries through the pre-collecting lymphatics to the collecting lymphatic vessels [4,5,7,10,12,15,16,17,18,19,20]. Pre-collecting lymphatics display sporadic intraluminal valves and a discontinuous vascular smooth muscle cell layer and basal lamina [21]. Collecting lymphatic vessels have intraluminal valves and are covered by a continuous vascular smooth muscle cell layer and basal lamina in most tissues. In contrast to the initial lymphatics, LECs of the collecting lymphatic vessels are interconnected with continuous cell junctions (zipper-like junctions), resulting in a reduced uptake of fluid, molecules, and cells from the interstitium [22]. Pulsative contractions of the vascular smooth muscle cells and the activity of surrounding skeletal muscles are the main driving factors of lymph flow, while intraluminal valves of pre-collecting and collecting lymphatics prevent backflow [23,24,25,26,27]. Lymph is propelled from the collecting lymphatic vessels into the subclavian veins through the thoracic duct or the right lymphatic trunk [1,2,3,4,5,6,7,10,12,15,16,17,18,19,20,28,29].

In association with the well-known roles of the lymphatic system in maintaining body fluid balance, cell uptake and trafficking, and dietary lipid absorption, lymphatic function contributes to the orchestration of inflammation and immune responses, tumor cell dissemination, and metastasis formation, while lymphatic malfunction has a central role in lymphedema formation [1,2,3,4,5,6,7,15,16,18,20,30,31,32,33,34,35]. Importantly, recent studies revealed the importance of organ-specific lymphatic function in a great variety of conditions, including cardiovascular diseases, obesity, or diseases affecting the central nervous system [20,36,37]. These recent findings emphasize a need for better understanding of how flow-generated mechanical forces contribute to the development and function of the lymphatic system.

## 2. Importance of Flow-Induced Mechanical Forces in Lymphatics

Experimental data demonstrated that ECs sense flow magnitude, direction, amplitude, and frequency of pulsatile flow [38,39,40,41,42,43,44]. A pulsatile, high-pressure flow is characteristic of the arterial system, while venous flow has low-pressure laminar nature, and low-pressure oscillatory flow is observed in lymphatic vessels. Lymphatic flow velocity and shear stress (the force per area acting on the vessel wall) rates [45,46,47] are below the values measured in blood vessels [48,49,50,51,52]. Although many studies focus on the effects of fluid shear stress and flow characteristics on blood vessels and BECs [48,53,54,55,56,57,58,59], the molecular interactions involved in shear stress mechanosensation and mechanotransduction are still poorly characterized in these structures. Our understanding on the molecular mechanisms involved in flow-induced signaling in LECs is even more limited [60].

### 2.1. Flow-Induced Cellular Changes in ECs

Previous in vitro experiments demonstrated that LECs and BECs react differently to the same mechanical conditions [61,62]. Under 4 dynes/cm^2^ planar shear, both LECs and BECs elongate parallel to the flow direction, while under interstitial flow with 10 μm/s average velocity, LECs form large vacuoles and exhibit long dendritic extensions in contrast to BECs that form branching, lumenized structures [61]. Prior in vitro experiments reported that acute shear stress induces a Ca^2+^-dependent reorganization of the cytoskeleton in BECs [63,64,65,66,67,68]. Dynamic reorganization of focal adhesion sites was observed in BECs upon acute shear stress that also supports flow-dependent cytoskeleton reorganization and cell alignment [69,70]. Another in vitro study demonstrated that although under 4 dynes/cm^2^ laminar shear stress, LECs become elongated and aligned with flow, LECs are more cuboidal under 4 dynes/cm^2^, ¼ Hz oscillatory shear stress (OSS), and their alignment is less dependent on flow direction [71]. The authors also revealed that flow-induced cytoskeleton reorganization and cell alignment of LECs are mediated by transcription factors forkhead box protein C2 (FOXC2) and prospero homeobox protein 1 (PROX1) [71]. In a recent in vivo study, FAT tumor suppressor homolog 4 (FAT4), encoded by the GATA binding protein 2 (GATA2) target gene *Fat4*, was identified as a key player in shear stress-dependent polarization of LECs [72].

An in vitro study revealed that slow interstitial flow with an average velocity of 4.2 μm/s synergizes with molecular factors promoting blood and lymphatic vessels sprouting by enhancing the availability of matrix-bound growth factors to the cells [73]. Similarly, in a recent study, it was demonstrated in a 3D in vitro model that interstitial flow with 1 μm/s average velocity augments the effects of pro-lymphangiogenic molecular factors and determines the direction of lymphatic sprouting [74].

According to these results, flow characteristics affect LEC shape, alignment, and sprouting, and these findings propose that fluid pressure and flow-generated mechanical forces may play a critical role in the development and function of lymphatic vessels.

### 2.2. Mechanical Forces in the Developmental Program of the Lymphatic Vasculature

#### 2.2.1. Early Steps of Lymphatic Development

The development of the lymphatic system begins between the 6th and 7th week of the 40-week-long pregnancy in humans [5] and approximately on the 9th embryonic day (E9.0) of the 21-day-long pregnancy in mice, when the first PROX1–lymphatic vessel endothelial hyaluronan receptor 1 (LYVE-1) double positive lymphatic progenitor cells bud from the cardinal vein and form the jugular lymph sac [75,76,77,78,79,80,81,82,83,84,85]. In mice, vascular endothelial growth factor receptor 3 (VEGFR-3) expression becomes restricted to the LECs from E10.5 [86,87,88]. In parallel, these cells start to express podoplanin (PDPN (other common aliases: T1α, gp38, E11 antigen)) [89,90]. A recent study revealed the role of GATA2 in VEGFR3 upregulation of LECs, which is critical for the directed VEGF-C-dependent migration of LECs during the early lymphatic development [91]. Immature pre-lymphatic vessels that are comprised of LECs carrying the molecular markers LYVE-1, PROX1, VEGFR-3, and PDPN [84,86,88,89,90] enmesh the mouse embryos [17].

Although LECs are predominantly derived from the venous endothelium [75,77,78,79,82,83,85,92,93,94,95,96], experimental data suggest multiple possible origins of LECs. Although the concept of the non-venous origin of lymphatics was first introduced by Huntington and McClure [97], it was neglected until recently. Almost a century later, mesenchymal lymphangioblasts were suggested to contribute to the development of avian and amphibian lymphatics in addition to venous ECs [98,99,100,101,102]. A subpopulation of LECs co-expressing mesenchymal-specific (cluster of differentiation 45 (CD45)) and LEC-specific (LYVE-1, PROX-1) markers was also observed in mouse embryos [103,104]. Moreover, a hemogenic-derived subpopulation of LECs was identified in the lymphatic vessels of the mesentery [105], dermis [106], and the heart [107]. Interestingly, recent findings suggest the contribution of second heart field progenitors to the development of cardiac lymphatics [108,109]. Organ-specific characteristics of lymphatic development and the origin of LECs were summarized in multiple reviews [9,10,11,37,110,111].

Interstitial fluid volume affects elongation and proliferation of LECs, presumably due to a mechanical force-induced activation of VEGFR-3 mediated by ß1 integrin [112], and regulated by integrin-linked kinase (ILK) signaling [113]. Other studies reported the role of interstitial flow in lymphatic vessel regeneration [114,115,116]. These findings suggest the possible role of interstitial fluid pressure-related mechanical forces in lymphatic expansion. Nevertheless, this proposed role of mechanical forces in the proliferation and migration of LECs is yet to be investigated.

#### 2.2.2. Common Steps of the Maturation of the Lymphatics

Further development and maturation of the lymphatic vessels take place in an organ-specific time and manner. During these processes, the immature pre-lymphatic plexuses connect to each other, additional lymphatic vessels are formed, and immature pre-lymphatic vessels undergo a structural remodeling that eventually leads to the formation of the initial lymphatic capillaries and collecting lymphatic vessels. Notably, mechanisms that regulate the maturation of lymphatic vessels are still poorly characterized. Most collecting lymphatics acquire a smooth muscle coverage, and lymphatic valves are formed in their lumen [17,117,118,119]. Numerous molecules and pathways, such as ephrin B2 [117] and B4 [120], angiopoietin 2 [121], integrin α9 [24], transcription factor nuclear factor of activated T cells 1 (NFATc1) [25,71], connexin 37 and 43 [71,122], bone morphogenetic protein (BMP-9) [123], mechanically induced Wnt/ß-catenin signaling [124], RAS p21 protein activator 1 (RASA1) [125], VEGFR-3 signaling [126,127] FAT4 [72,128], platelet endothelial cell adhesion molecule (PECAM) [129], vascular endothelial cadherin (VE-cadherin) [130], neuropilins, semaphorins, and plexins, [131,132] have been revealed to contribute to the development of the intraluminal lymphatic vessel valves. Recent data suggest that yes-associated protein (YAP)/transcriptional coactivator with PDZ-binding motif (TAZ) signaling might be critical for the maturation and maintenance of lymphatic valves [133,134]. Transcription factors GATA2 [135], FOXC2 [25,136], and PROX1 [71] were identified as key molecular factors of the lymphatic valve development, and their expression is also critical for the maintenance of these structures [137,138,139]. GATA2 and FOXC2 are also essential for the establishment of a proper vascular smooth muscle coverage of the collecting lymphatic vessels [136,138]. In primary LECs isolated from the dermis of E16.5 embryonic mice, GATA2 has been demonstrated to regulate the expression of numerous genes encoding proteins involved in lymphatic valve formation, including FOXC2, PROX1, NFATc1, PECAM, angiopoietin 2, or integrin α9 [135].

#### 2.2.3. Separation of the Blood and Lymphatic Systems

As mentioned above, the blood vasculature and the lymphatic vessel systems are connected in the jugular region where the thoracic duct and the right lymphatic trunk enter the subclavian veins. Two parallel mechanisms are responsible for the prevention of backflow of blood into the lymphatic system. Lymphovenous valves (LVV) develop between the jugular lymph sac and the cardinal vein at E11.5–E13.5 in mice (Figure 1A,B) [140,141]. Flow-generated mechanical forces and molecular factors that are involved in lymphatic valve formation, such as FOXC2, GATA2, PROX1, integrin α9, connexin 37, and Wnt/ß-catenin signaling, were shown to contribute to LVV morphogenesis [124,138,140,141,142]. The role of YAP/TAZ signaling in the development and maintenance of LVVs is also suggested by a recent study [134].

Besides LVV formation, another mechanism is required to ensure the separation of the blood and lymphatic systems. C-type lectin-like receptor 2 (CLEC-2), which is involved in the spleen tyrosine kinase (SYK)-dependent platelet activation [143,144], is a receptor for PDPN expressed on the surface of LECs [144,145,146], and PDPN activates platelets by the CLEC-2/SYK/SH2-domain-containing leukocyte protein of 76 kDa (SLP-76)/phospholipase C gamma 2 (PLCγ2) pathway [147]. This platelet-mediated molecular pathway is essential for the appropriate separation of the blood and lymphatic circulations (Figure 1C) [89,144,147,148,149,150,151,152,153,154,155,156,157,158,159].

#### 2.2.4. Lymph Flow Promotes Organ-Specific Maturation of the Lymphatic Vasculature

Lymph flow is reportedly reduced in mice with impaired PDPN/CLEC-2/PLCγ2 signaling [160,161,162] due to an improper separation of the blood and lymphatic vessel systems [156]. These data propose that these genetic mouse models provide ideal tools to study the roles of lymphatic flow in vivo. Importantly, while the formation of immature mesenteric pre-lymphatics is intact in mice with impaired lymph flow due to a lack of CLEC-2, organ-specific maturation of the mesenteric lymphatic vessels is impaired in these embryos [160], and the phenotype is also present in PLCγ2-deficient embryos [162]. In addition, the postnatal developmental program of the meningeal lymphatic vasculature is also affected in PLCγ2-deficient mice [162]. These in vivo findings underline the importance of mechanical forces in the organ-specific developmental program of the lymphatic vessels.

Taken together, flow-induced mechanical forces regulate LEC shape and alignment, promote lymphatic sprouting and development, and contribute to the maturation of the lymphatic vessel systems. These findings suggest that LECs bear mechanisms responsible for mechanosensation and mechanotransduction that make them able to sense the surrounding mechanical forces and translate these factors to molecular levels, leading to regulation of gene expression, cell proliferation, survival, migration, and identity, in addition to cytoskeleton remodeling.

### 2.3. Molecular Mechanisms Involved in the Mechanosensation and Mechanotransduction of LECs

#### 2.3.1. Shear Stress-Dependent Molecular Pathways in Lymphatic Valve Morphogenesis

Intraluminal valves develop after the increase in lymphatic flow, and it has been demonstrated that OSS (either 4 dynes/cm^2^, ¼ Hz or an average 0.67 dynes/cm^2^ with a maximum of 3.25 dynes/cm^2^ and a minimum of −1.25 dynes/cm^2^, 1 Hz) upregulates the expression of transcription factors FOXC2, GATA2 [71,138,160], and their downstream molecular factors, such as connexin 37, integrin α9, ephrin B2, neuropilin 1, and Krüppel-like factor 2 (KLF-2) [71,138,160], presumably in a VE-cadherin/Wnt/ß-catenin dependent manner [124,130,163]. In these in vitro experiments, PROX1 levels were not altered by OSS [71,160]. Nevertheless, other in vitro data suggest that GATA2 promotes *PROX1* transcription [135,138]. Importantly, lymphatic valves are formed mainly at lymphatic branches [24,25,71], where the laminar nature of the flow is interrupted (Figure 2) [53,164,165]. Taken together, these findings further promote the hypothesis that the lymph flow dynamics and local shear stress characteristics play an important role in the development of the intraluminal lymphatic valves. Besides the molecular pathways mentioned before, the role of syndecan 4 was identified in the sensation of flow direction [166] and lymphatic valve formation [129]. However, the mechanism of its flow-dependent signaling that promotes lymphatic development is still poorly understood.

#### 2.3.2. Mechanical Forces as Possible Regulators of Spatial Changes in Gene Expression during the Maturation of Lymphatic Vessels

Expression of PROX1, FOXC2, and VEGFR-3 remains high in the lymphatic valve cells but is reduced in the mature lymphatic vessel walls after the formation of lymphatic valves [25], presumably due to changes in flow patterns downstream of lymphatic valves after their development (Figure 2B). The role of steady shear flow-induced epsin signaling in the temporal and spatial regulation of VEGFR-3 abundance in LECs was demonstrated [126], and the mechanism may contribute to the mechanical force-dependent site-specific changes in gene expression in LECs.

Recent studies suggest the possible role of YAP/TAZ signaling in flow-dependent regulation of gene transcription in LECs. Transcription factor TAZ is expressed in lymphatic valve forming LECs [133,134,137], and in vitro data suggest that 4 dynes/cm^2^, ¼ Hz OSS enhances the nuclear translocation of YAP and TAZ, which is necessary for their function in regulating gene transcription [137]. Some investigators in their in vitro and in vivo experiments found that VEGF-C signaling is also capable of inducing the YAP/TAZ pathway that in turn enhances the expression of PROX1 [134]. However, other researchers reported that YAP/TAZ signaling is negatively regulated by the VEGF-C–VEGFR-3 pathway and represses PROX1 expression [133]. In this study, it was also observed that LEC-specific deletion of YAP and TAZ inhibits lymphatic branch formation and proper lymphatic development. Moreover, enrichment of PROX1 in LV forming LECs and downregulation of PROX1 in luminal LECs was also repressed in this model [133]. Importantly, both studies support that YAP/TAZ signaling promotes the PROX1–VEGFR-3 pro-lymphangiogenic positive feedback loop. Another study using in vitro and in vivo approaches demonstrated that FOXC2 inhibits the downstream signaling of the YAP/TAZ pathway, thereby suppressing its pro-proliferative effects [137], which may serve as an offset against the PROX1–VEGFR-3 positive feedback loop. FOXC2-dependent repression of shear stress-induced YAP/TAZ signaling may be a potential mechanism answering why PROX1 expression levels were unaltered under OSS in in vitro experiments [71,160], although GATA2 promotes the transcription of *PROX1* and *FOXC2* according to other in vitro experimental results [135,138]. These findings propose that YAP/TAZ signaling influenced by both local shear stress characteristics and molecular factors may play a central role in orchestrating the spatial gene expression patterns in lymphatic valve-forming and luminal LECs. Nevertheless, roles of the YAP/TAZ pathway and the underlying mechanisms of its signaling are still barely understood in LECs.

#### 2.3.3. Ion Channels in Shear Stress-Dependent Intracellular Signaling Mechanisms

Activation of these shear stress-induced transcription factors requires mechanosensory mechanisms that regulate gene expression and cellular functions. Recent studies revealed the role of piezo-type mechanosensitive ion channel component 1 (PIEZO1)-mediated mechanotransduction in the development and maintenance of the lymphatic valves [167,168]. EC-specific deletion of *PIEZO1* in mice results in the absence of lymphatic valves, chylous pleural effusion, and postnatal mortality due to impaired lymphatic function [167]. Post-developmental LEC-specific deletion of *PIEZO1* leads to a significant degeneration of both lymphatic valves and lymphatic vessels in the skin and mesentery [168]. In humans, absence or loss of function mutations of the *PIEZO1* gene causes hereditary lymphoedema as result of defects in the lymphatic vessel developmental program [169,170]. PIEZO1 is a pore-forming subunit of mechanically activated cation channels [171,172,173] with a rapid inactivation rate, suggesting its role as a sensor of transient stress [174]. PIEZO1 presumably contributes to shear stress-mediated development of lymphatic vessels by increasing intracellular Ca^2+^ levels in LECs, similarly to BECs, in which it has been demonstrated that acute shear stress elicits a rise in intracellular Ca^2+^ levels from intracellular stores and activation of Ca^2+^ channels [175,176] and PIEZO1 is essential for shear-stress dependent vascular development [177,178,179].

Importantly, PIEZO1 is not the only mechanosensitive ion channel that has been proposed to affect the maturation of the lymphatic vessels. Calcium release-activated calcium modulator 1 (ORAI1), a pore subunit of the calcium release-activated calcium (CRAC) channel, is activated upon shear stress and mediates Ca^2+^-influx in LECs [180]. Laminar flow induces an ORAI1-dependent upregulation of KLF-2 and KLF-4 in LECs that promote VEGF-C expression, among other molecular factors contributing to the cell cycle progression [181]. VEGF-C signaling is reportedly regulated by sphingosine-1-phosphate receptor 1 (S1PR1) in quiescent LECs [182]. In addition to their role during the development of lymphatic vessels, CRAC channels are also proposed to mediate Ca^2+^-dependent regulation of lymphatic barrier function [183].

Intracellular Ca^2+^, acting together with KLF-2 and PROX1, is suggested to play a central role in shear stress-induced lymphatic sprouting and development [180,181]. However, in vitro data suggest that OSS (an average 0.67 dynes/cm^2^ with a maximum of 3.25 dynes/cm^2^ and a minimum of −1.25 dynes/cm^2^, 1 Hz) induces the expression of KLF-2 but not PROX1 [160], which further supports the role of KLF-2 in lymphatic shear stress-induced signal transduction.

#### 2.3.4. The PECAM–VE-Cadherin–VEGFR-2/3 Complex as a Potential Mechanoreceptor Complex in Lymphatic Endothelial Cells

PECAM, VE-cadherin, VEGFR-2, and VEGFR-3 form a mechanosensory complex in ECs [184,185], which is proposed to promote cell proliferation by activating the phosphatidylinositol-3-kinase (PI3K)/Akt pathway and cytoskeleton remodeling in a flow-dependent manner [185,186,187,188,189].

Pan-endothelial marker PECAM is assumed to play an upstream transmitting role in this complex [185]. However, the mechanisms that evoke a flow-dependent activation of PECAM are still poorly understood [190]. It has been revealed previously that PECAM-deficiency results in an impairment of flow response of blood vessels and inhibits arteriogenesis and vascular remodeling in mice [187,191,192]. Moreover, PECAM was also demonstrated to play a role in lymphatic mechanosensation and valve formation during the embryonic maturation of lymphatic vessels, as PECAM-null mouse embryos display lymphatic remodeling defects [129].

VE-cadherin functions as an adaptor by binding to VEGFR-2 and VEGFR-3 [184,185,189], and its distribution within the cell membrane depends on the spatial flow characteristics [193]. Importantly, LEC-specific conditional deletion of VE-cadherin results in lymphatic valve defects in mice [130].

VEGFR-2 is known for playing various important roles in the cardiovascular system [194]. VEGFR-2-related ligand-independent activation of PI3K/Akt and Erk pathways in response to shear stress promotes cell proliferation, contributes to cytoskeletal remodeling, and impairs junctional remodeling and stabilization in BECs [185,195,196,197,198,199,200,201]. In addition, VEGFR-2 has been shown to be expressed on LECs [202] and is suggested to support VEGFR-3 signaling in the lymphatic developmental program and adult lymphangiogenesis [202,203,204].

As discussed above, flow-induced mechanical forces promote the sprouting and development of lymphatic vessels and play an important role in the stabilization of mature lymphatics. Flow-induced mechanical forces also contribute to similar processes in the blood vasculature [205,206,207,208]. Importantly, ECs with different cell fate commitments react differently to the same mechanical conditions [61,62]. Flow characteristics vary greatly among different vessel types [45,46,47,48,49,50,51,52], and changes in the nature of flow are sufficient to re-program the identity of differentiated ECs [159,180,209]. Therefore, a mechanism that determines the appropriate shear stress sensitivity of the ECs is essential for the proper organization of the vascular systems. It has been demonstrated that the shear sensitivity of the ECs is largely dependent on the VEGFR-3 expression of the cells [44]. Overexpression of VEGFR-3 in BECs significantly increased their sensitivity to shear stress in vitro, which suggests that ECs with a higher VEGFR-3 expression are more sensitive to shear stress due to a lower shear stress set point. This is in line with the phenomenon that VEGFR-3 expression is restricted to lymphatic vessels [86,87,88] exposed to lower flow velocity and shear stress levels compared to those in arteries or veins [45,46,47,48,49,50,51,52].

Interestingly, in addition to its well-known roles in lymphatic vessel development and lymphangiogenesis [4,5,7,15,16,17], VEGFR-3 expression was demonstrated in the endothelium of the aorta [44,184], Schlemm’s canal [210,211,212], ascending *vasa recta* [213], and the spiral arteries [214] in mice. Importantly, the role of VEGF-C – VEGFR-3 signaling in the development of Schlemm’s canal and the remodeling of the spiral arteries was demonstrated [211,212,214]. VEGFR-3 was previously considered to be primarily expressed on LECs [86,87,88], promoting lymphatic commitment of the ECs [82,215,216].

In vitro results suggest that the expression of VEGFR-3 on BECs is dependent on fluid shear stress characteristics [217]. Importantly, in a mature lymphatic network, VEGFR-3 expression also differs among LECs exposed to different local flow characteristics [25,141]. These data suggest that VEGFR-3 expression in ECs is regulated, at least in part by shear stress-induced mechanisms, and VEGFR-3 expression levels may affect the shear stress sensitivity of the PECAM–VE-cadherin–VEGFR-2/3 mechanosensory complex by a still uncovered mechanism. Moreover, VEGFR-3 signaling may have a still poorly characterized function in regulating endothelial cell plasticity, which may be in relation to its presumed role in determining shear stress sensitivity of endothelial cells.

Although the PECAM–VE-cadherin–VEGFR-2/3 mechanoreceptor complex has been investigated in BECs, the results indicating the involvement of PECAM [129] and VE-cadherin [130] in lymphatic valve formation and lymphatic remodeling, in combination with the well-known functions of VEGFR-3 in lymphatic development, suggest that these molecular factors may also contribute to these mechanical force-dependent processes in LECs.

## 3. Closing Remarks

Taken together, the presented findings support that the flow-induced mechanical force dynamics play critical roles in the determination and maintenance of EC fate; regulate LECs shape and alignment and maturation of the lymphatic network; promote lymphatic sprouting and development; and orchestrate the morphogenesis and maintenance of the lymphatic valves and the lymphovenous valve.

Mutations affecting genes that contribute to the maturation of lymphatic vessels often result in diseases associated with lymphatic malfunction in clinical studies, such as *GATA2* (primary lymphedema with myelodysplasia progressing to acute myeloid leukemia) [135,218,219], *FOXC2* (lymphedema distichiasis) [220,221,222], *FAT4* (Hennekam syndrome with no mutations in collagen and calcium binding EGF domains 1 gene (*CCBE1*) [223], *ITGA9* encoding integrin α9 (fetal severe chylothorax) [224], *PIEZO1* (autosomal recessive form of generalized lymphatic dysplasia) [170], *FLT4* encoding VEGFR-3 (Milroy’s disease) [225,226,227], and *EPHB4* encoding ephrin B4 (lymphatic-related hydrops fetalis) [120]. As most of these molecular factors are proposed to be regulated at least in part by flow-induced mechanical forces, these data emphasize the possible clinical relevance of lymph flow-induced molecular pathways. Despite the recent important advancements in understanding the mechanisms of mechanosensation and mechanotransduction in LECs (Figure 3), the molecular background of these processes is still poorly understood.

A better understanding of molecular pathways and interactions that are involved in the mechanical force-related signaling of LECs is needed. Uncovering the molecular background of flow-dependent mechanisms in lymphatic vessels may lead to identification of biomarkers improving the diagnosis of lymphatic diseases or may provide novel therapeutic targets for pathological conditions related to a dysfunction of lymphatics, such as lymphedema.

## Figures and Tables

**Figure 1 ijms-22-03955-f001:**
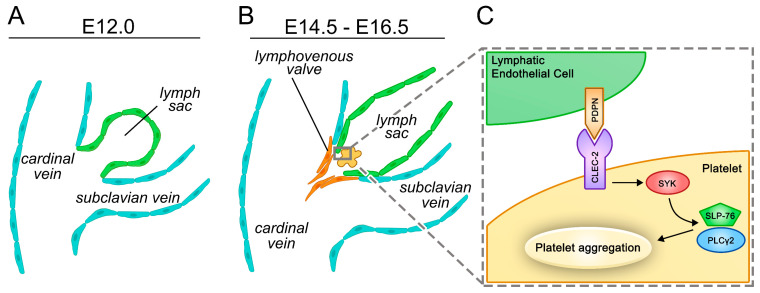
Cellular and molecular mechanisms involved in the separation of the blood and lymphatic vessel systems in mice. (**A**,**B**) Overview of the separation of the cardinal vein and the lymph sac with the formation of the lymphovenous valve in parallel with the platelet-dependent activation of the C-type lectin-like receptor 2 (CLEC-2)/spleen tyrosine kinase (SYK)/SH2-domain-containing leukocyte protein of 76 kDa (SLP-76)/phospholipase C gamma 2 (PLCγ2) pathway. (**C**) The CLEC-2/SYK/SLP-76/PLCγ2 pathway in platelets recognizing podoplanin (PDPN) expressed on the surface of lymphatic endothelial cells (LECs) is critical for the proper separation of the two circulatory systems.

**Figure 2 ijms-22-03955-f002:**
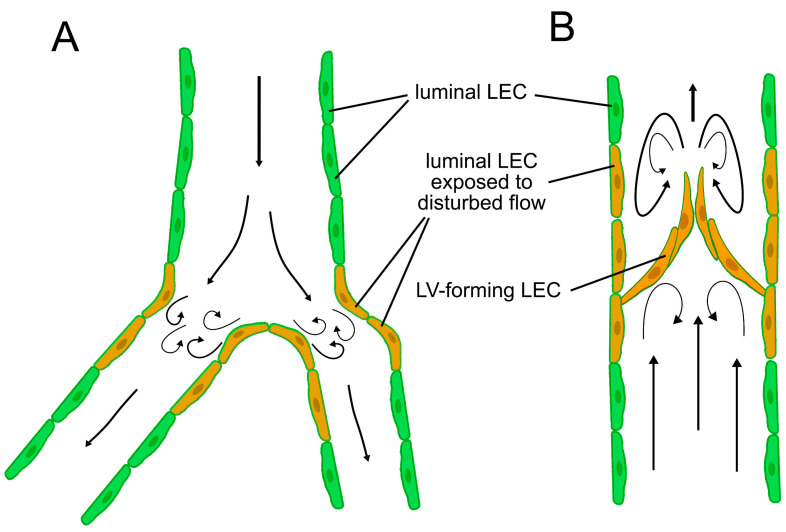
Flow patterns in the lymphatic system. (**A**) LECs in branches are exposed to disturbed flow patterns with higher shear stress even before the establishment of intraluminal valves. (**B**) Flow becomes disturbed at the lymphatic valves of mature collecting lymphatic vessels. Valve-forming LECs and luminal LECs surrounding lymphatic valves are also exposed to disturbed flow, while LECs at other sites than branches, valves, and curvatures are exposed to a more laminar flow.

**Figure 3 ijms-22-03955-f003:**
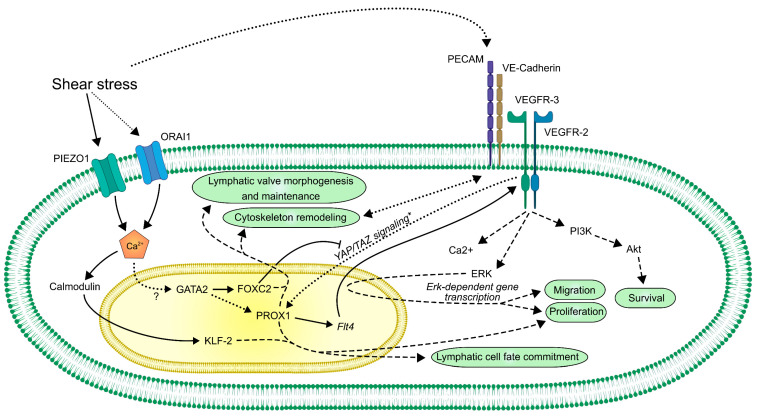
A schematic overview of the currently known mechanosensory and mechanotransduction molecular pathways in LECs. Lines represent direct connections between molecules, such as regulation of transcription, complex formation, direct activation, or inactivation. Dashed lines represent an indirect connection. Dotted lines represent assumed or still unclear connections.

## Data Availability

Not applicable.

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
