# Peer review of "Mechanosensation and Mechanotransduction by Lymphatic Endothelial Cells Act as Important Regulators of Lymphatic Development and Function"

_ijms, 2021, doi:10.3390/ijms22083955_

Round 1

Reviewer 1 Report

In the present review article “Mechanosensation and mechanotransduction by lymphatic endothelial cells act as important regulators of lymphatic development and function” Bálint and Jakus summarize the effects of mechanical forces on lymphatic vessel morphogenesis and regulation of various lymphangiogenic signaling mechanisms. Furthermore, they discuss the molecular mechanisms in mechanosensing and mechanotransduction in lymphatic endothelial cells. The review article is well-written and timely. However, I have the following minor comments.

  1. There are many grammatical errors and language needs editing.
  2. It is suggested to include the information about the organ-specific origin of lymphatic vessels in subheading 2.2.1 as discussed in the article by Petrova and Koh (PMID 29242199). Lineage-tracing experiments suggested an increased variability in lymphatic endothelial cells' origin in different organs.

Author Response

We would like to thank the Reviewer for his/her comments and appreciate the suggestions.

We revised the manuscript text and corrected the grammatical errors. We also reformulated some sentences to improve their clarity.

We appreciate the suggestion on the importance of the organ-specific origins of lymphatic vessels. We included a new paragraph in the revised manuscript discussing the possible origins of LECs, and referenced excellent recent review articles on the topic.

We hope that the Reviewer finds the manuscript acceptable in its current form.

Reviewer 2 Report

The review article by L. Balint and Z. Jakus aims to focus on lymphatic development and function, in particular the mechanical forces contributing to the endothelial cell (EC) fate commitment and the processes of maturation of the lymphatic network.

In this review authors provide an interesting overview of the various effects of mechanical forces on lymphatics and summarize the current understanding of the molecular bases involved in the mechanosensation and mechanotransduction in lymphatic endothelial cells.

They describe morphology of the lymphatic EC (LEC) in the lymphatic vasculature and the roles of flow-induced forces in lymphatic vessels.

Then they analyze early steps of lymphatic development, the common steps of the maturation of the lymphatic vessels, the separation of the blood and lymphatic systems, as well as the mechanosensation and mechanotransduction mechanisms in LEC.

The article comes across as well written and well coordinated in its parts. English written is good.

However, the authors only mention the main pathology of LEC, i.e. lymphoedema. They could also mention some other pathologies such as lymphadenitis due to infections or carcinoma, whereas primary lymphangiosarcoma is very rare.

Therefore, in my opinion, it would be necessary to implement some data on LEC and metastasis dissemination, because this point is of extreme interest for global health.

Author Response

We thank the Reviewer for the kind comments and the valuable suggestions.

We completely agree that the role of lymphatic vessels in tumor dissemination is of great interest. We expanded the corresponding part of the manuscript with this important function. We also included more details about the other pathological functions of the lymphatics. However, it is important to note that it is limited what can be discussed in detail in the current manuscript. Importantly,
excellent prior review articles summarized these topics, which we reference in the current form of the manuscript. Here we wanted to give a brief summarization of the common pathological functions of the lymphatics, and keep the manuscript well focused.

We hope that this is acceptable for the Reviewer.
